# m^6^A Methylation Regulators Are Predictive Biomarkers for Tumour Metastasis in Prostate Cancer

**DOI:** 10.3390/cancers14164035

**Published:** 2022-08-21

**Authors:** Yingchun Liang, Xiaohua Zhang, Chenkai Ma, Jimeng Hu

**Affiliations:** 1Department of Urology, Huashan Hospital, Fudan University, No. 12 WuLuMuQi Middle Road, Shanghai 200040, China; 2Fudan Institute of Urology, Huashan Hospital, Fudan University, Shanghai 200040, China; 3Molecular Diagnostic Solution, Nutrition and Health, Health and Biosecurity, CSIRO, North Ryde 2113, Australia

**Keywords:** prostate cancer, metastasis, biomarker, m^6^A, methylation, immune cell infiltration

## Abstract

**Simple Summary:**

Recurrence and metastatic progression always lead to dismal outcomes in prostate cancer (PCa). There is no reliable biomarker for the prediction of recurrence and metastasis other than the Prostate Cancer Antigen (PCA). N6-methyladenosine (m^6^A) is the most common post-transcriptional mRNA modification and is regulated by m^6^A regulators dynamically. Since m^6^A modification is associated with cancer development and outgrowth, we performed a consensus clustering on PCa with regard to the gene expression of all m^6^A regulators. We identified three subtypes of Pca with distinct m^6^A expression patterns and enriched biological pathways. We also established an m^6^A score for metastasis prediction based on our clustering, which is potentially a predictive biomarker for Pca metastasis.

**Abstract:**

Prostate cancer (PCa) is one of the most common cancers in men. Usually, most PCas at initial diagnosis are localized and hormone-dependent, and grow slowly. Patients with localized PCas have a nearly 100% 5-year survival rate; however, the 5-year survival rate of metastatic or progressive PCa is still dismal. N6-methyladenosine (m^6^A) is the most common post-transcriptional mRNA modification and is dynamically regulated by m^6^A regulators. A few studies have shown that the abnormal expression of m^6^A regulators is significantly associated with cancer progression and immune cell infiltration, but the roles of these regulators in PCa remain unclear. Here, we examined the expression profiles and methylation levels of 21 m^6^A regulators across the Cancer Genome Atlas (TCGA), 495 PCas by consensus clustering, and correlated the expression of m^6^A regulators with PCa progression and immune cell infiltration. Consensus clustering was applied for subtyping Pca samples into clusters based on the expression profiles of m^6^A regulators. Each subtype’s signature genes were obtained by a pairwise differential expression analysis. Featured pathways of m^6^A subtypes were predicted by Gene Ontology. The m^6^A score was developed to predict m^6^A activation. The association of the m^6^A score with patients’ survival, metastasis and immune cell infiltration was also investigated. We identified three distinct clusters in PCa based on the expression profiles of 21 m^6^A regulators by consensus clustering. The differential expression and pathway analyses on the three clusters uncovered the m^6^A regulators involved in metabolic processes and immune responses in PCa. Moreover, we developed an m^6^A score to evaluate the m^6^A regulator activation for PCa. The m^6^A score is significantly associated with Gleason scores and metastasis in PCa. The predictive capacity of the m^6^A score on PCa metastasis was also validated in another independent cohort with an area under the curve of 89.5%. Hence, our study revealed the critical role of m^6^A regulators in PCa progression and the m^6^A score is a promising predictive biomarker for PCa metastasis.

## 1. Introduction

Prostate cancer (PCa) is the most common tumor in men. Although PCa is usually benign and localized in the initial diagnosis, metastasis is still the most frequent cause for prostate cancer-related mortality [1,2]. Metastatic PCa is responsible for more than 90% of deaths in patients with PCa. The rapid growth and progression of metastatic PCa is largely driven by androgen; therefore, androgen-deprivation therapy (ADT) is one of the first-line treatments for symptomatic metastatic PCa [3]. However, the duration of response of ADT is variable, and resistance is inevitable in metastatic prostate cancer after ADT. The outcomes for metastatic castration-resistant prostate cancer (mCRPC) are still dismal, though a few attempts including new algorithms using MRI and new regimens have been investigated to improve their efficacy for mCRPC [4,5,6,7,8,9]. Despite the significant accuracy of predicting a positive biopsy for PCa, little is known about the biological markers on PCa. Therefore, a further understanding of the driving force and regulatory molecules in mCRPC is key to improving the outcomes of patients with mCRPC.

Previous studies have uncovered a few potential mechanisms of resistance in mCRPC, independent of driver mutation status [10]. ADT impairs androgen receptor (AR) signaling-dependent cell growth in prostate cancer by the reduction of secretory androgen [11]. However, the AR signaling is likely restored after a few periods of ADT. In addition, ADT can introduce the bypass of AR signaling by converting adrenal precursors into testosterone along with alternative activation of other receptor-signaling pathways, such as the PTEN/PI3K/Akt pathway and DNA damage repair pathway [12]. AR-independent signaling also highlights the heterogeneity of mCRPC with regard to gene expression and signaling transduction [13]. Moreover, neuroendocrine differentiation is another critical phenotype resulting in ADT resistance, which leads to the lack of AR expression on the surface of PCa cells [14]. Hence, further investigation on the regulatory mechanism beyond AR signaling and the relationship of heterogeneity and the metastatic phenotype in PCa will precisely stratify patients according to their metastatic risk.

N6-methyladenosine (m^6^A) is the most prominent RNA modification regulating the transcription, stabilization and translocation of RNA without DNA or RNA base changes [15]. Similar to DNA methylation, m^6^A is a biologically reversible process that is elaborately regulated by methyltransferases, demethylases and binding proteins, also known as “writers”, “erasers” and “readers”, respectively [16]. The methylation marks of m^6^A are dynamically catalyzed by methyltransferases, usually consisting of RBM15, ZC3H13, METTL3, METTL14 and WTAP, and removed by demethylases, such as FTO and ALKBH5 [17]. The RNA-binding proteins (“readers”) YTHDF1/2/3, YTHDC1/2, HNRNPA2B1, LRPPRC and FMR1 can recognize the m^6^A status in RNAs and mediate the biological functions of these RNAs [18]. Accumulating evidence has demonstrated that m^6^A modification is involved in multiple biological processes. Neural-simulated memory learning is facilitated by m^6^A regulators [19,20]. A decrease in pluripotency is also observed in stem cells with a high activation of m^6^A regulators [21,22,23,24]. Inactivation of m^6^A regulators is associated with cancer metastasis in the liver, colon, kidney and pancreas [17,25,26,27,28,29]. In PCa, high expression of METTL3 elevates the growth and progression of cancer cells [30]. Moreover, the subtypes of cancer stratified by m^6^A regulators have distinct prognoses, indicating that m^6^A regulators likely contribute to the heterogeneity of mCRPC [31] and the metastatic risk of Pca could be inferred by the expression levels of m^6^A regulators. However, the comprehensive m^6^A regulators expression pattern, along with their genetic alterations and DNA methylation patterns, and the predictive capacity of m^6^A for Pca metastasis remain unclear in Pca.

Here, we utilized the RNA sequencing data of 454 PCas to establish the comprehensive expression patterns of m^6^A regulators. We identified differentially expressed genes between subtypes with distinct levels of m^6^A regulators that were enriched in the immune response and metabolic regulation. Immune cell infiltration was also associated with m^6^A regulator expression patterns. For application of metastatic prediction by m^6^A regulator patterns, we constructed an m^6^A score. This m^6^A score demonstrated a moderate prediction for PCa metastasis. 

## 2. Materials and Methods

### 2.1. Data Pre-Processing

Public gene expression data and full clinical annotations of prostate adenocarcinoma (PRAD) were downloaded from the Cancer Genome Atlas (TCGA) database. RNA sequencing data (FPKM value) of gene expression and DNA methylation profiles by Illumina 450 K were downloaded from the Genomic Data Commons (GDC, https://portal.gdc.cancer.gov/ (accessed on 20 March 2020)). The mutation information with annotations of corresponding samples from TCGA PRAD were downloaded from Firehose (https://gdac.broadinstitute.org/ (accessed on 20 March 2020)). The CpG loci were annotated by the package “IlluminaHumanMethylation450kanno.ilmn12.hg19”. For gene expression analysis, batch effects from non-biological technical biases were removed using the “ComBat” algorithm of the “sva” package.

### 2.2. Consensus Clustering

An unsupervised consensus clustering according to the expression profiles of 21 m^6^A related genes was utilized to investigate the subtypes of PCa. The m^6^A regulators included 8 writers (METTL3, METTL14, RBM15, RBM15B, WTAP, KIAA1429, CBLL1, ZC3H13), 2 erasers (ALKBH5, FTO) and 11 readers (YTHDC1, YTHDC2, YTHDF1, YTHDF2, YTHDF3, IGF2BP1, HNRNPA2B1, HNRNPC, FMR1, LRPPRC, ELAVL1). First, we performed the Mann–Whithey U test to select the differentially expressed m^6^A regulators between PCa and normal prostate tissue. We only selected the significantly expressed m^6^A regulators with a *p* value of less than 0.05 for downstream analysis. Then, we applied agglomerative hierarchical clustering with average linkage to detect robust clusters in PCa. The distance metric was 1 (Pearson’s correlation coefficient) and was run over 1000 iterations. SigClust was performed to establish the significance of difference between each cluster in a pairwise matter. Silhouette width is defined as the ratio of each sample’s average distance to samples in the same cluster to the smallest distance to samples not in the same cluster. The best number of the cluster group is determined by the Cumulative Distribution Function (CDF) curve and the delta change of CDF in each subtype number.

### 2.3. Differentially Expressed Gene (DEG) Analysis

The differentially expressed genes between subtypes were identified by the “limma” package in a pairwise fashion. The significance was defined as log fold change larger than 2 and FDR less than 0.05. The list of DEGs was determined after removal of the replicated genes in each comparison.

### 2.4. Pathway Enrichment and Evaluating Immune Cell Abundance

The enriched functional pathway of DEG was annotated by Metascape (https://metascape.org/gp/index.html (accessed on 10 April 2020)) in the express module. *p*-values were calculated based on the cumulative hypergeometric distribution, while *q*-values were calculated using the Benjamini–Hochberg procedure to account for multiple testings. Pathway terms with *p*-values < 0.01, minimum counts of 3, and enrichment factors > 1.5 (the enrichment factor is the ratio between the observed counts and the counts expected by chance) were collected and grouped into clusters based on their membership similarities. PaGenBase was utilized to evaluate the tissue specificity of DEGs. The Gene Set Enrichment Analysis (GSEA) was conducted in GSEA 4.0.2 for identification of enriched pathways in a pairwise fashion. The immune cell infiltration in PCa was predicted by CIBERSORT (https://cibersort.stanford.edu/ (accessed on 10 April 2020)) with 100 permutation runs. 

### 2.5. M^6^A Score

To quantify the m^6^A regulator patterns of individual samples, we constructed a set of scoring systems to evaluate the m^6^A regulator levels of individual patients with PCa. The m^6^A score was established according to previous studies with some modifications [32]. In brief, first, we filtered out m^6^A regulators that were not differentially expressed between PCa and normal prostate tissue. Second, we utilized the Student’s *t* test to identify the m^6^A signatures associated with PCa metastasis and only took the differentially expressed m^6^A regulators between metastatic PCa and non-metastatic PCa into the establishment of the m^6^A score. We then performed principal component analysis (PCA) to construct the m^6^A regulator signature patterns. Principal components 1 and 2 were both selected to form the m^6^A score (Appendix A).
(1)$$m^{6}A\;\mathrm{score}\;=\overset{}{\underset{}{\sum }}PC1\;i+PC2\;i$$
where *i* is the PCa metastasis-related signature gene.

### 2.6. Stemness Index

The Stemness Index of each sample predicted by an OCLR algorithm has been described before. In brief, Malta et al. established a predictive model on pluripotent stem cell samples to train a stemness signature [33]. A gene expression profile containing 24 genes was included in the mRNA expression-based signature. Applying this model based on 24 stemness signatures, the stemness is referred to regardless of the tumor purity. We applied the stemness index model to score the PCa samples using RNA expression data. The Stemness Index was subsequently scaled to the [0, 1] range by subtracting the minimum and dividing by the maximum.

### 2.7. Statistical Analysis

Correlation coefficients between the m^6^A score and the expression of m^6^A regulators, as well as Stemness, were determined by Spearman correlation analyses. One-way ANOVA was used to conduct difference comparisons of three or more groups. The overall survival (OS) and recurrence-free survival (RFS) of each m^6^A subtype were compared to investigate the correlation of m^6^A subtypes and patient outcomes. The survival curves for the survival analysis were generated via the Kaplan–Meier method, and log-rank tests were utilized to identify the significance of differences between subtypes. The specificity and sensitivity of the m^6^A score and the gene SNPH were assessed through a receiver operating characteristic (ROC) curve. The area under the curve (AUC) was quantified and plotted using the pROC R package. The waterfall function of the maftools package was used to present the mutation landscape of all m^6^A regulators in patients with PCa from the TCGA cohort. For DNA methylation levels, the CpG loci located in promoter areas (genomic location: TSS500 or TSS1500) of m^6^A regulators were utilized for comparison between m^6^A subtypes. All statistical *p*-values were two-sided, with *p* < 0.05 being statistically significant. All data processing was performed in R 3.6.0 software (R Core Team, Vienna, Austria).

## 3. Results

### 3.1. Consensus Clustering of PCas Based on the Expression Profiles of m^6^A Regulators Suggested Three Biologically Distinct Subtypes in PCa

To investigate the similarity and discrepancy of m^6^A regulator expression patterns between PCas, we first performed consensus clustering on 495 PCa samples from the TCGA PRAD dataset using 21 pre-defined m^6^A regulators (METTL3, METTL14, RBM15, RBM15B, WTAP, KIAA1429, CBLL1, ZC3H13, ALKBH5, FTO, YTHDC1, YTHDC2, YTHDF1, YTHDF2, YTHDF3, IGF2BP1, HNRNPA2B1, HNRNPC, FMR1, LRPPRC, ELAVL1). First, 18 of 21 m^6^A regulators (METTL3, METTL14, RBM15B, KIAA1429, CBLL1, ZC3H13, ALKBH5, FTO, YTHDC1, YTHDC2, YTHDF1, YTHDF2, IGF2BP1, HNRNPA2B1, HNRNPC, FMR1, LRPPRC, ELAVL1) were identified as differentially expressed in PCa compared to normal prostate tissue (Appendix A), suggesting that most m^6^A regulators were biologically pro-active in PCa. Therefore, we selected these 18 m^6^A regulators as key m^6^A regulators for consensus clustering. Consensus average linkage hierarchical clustering identified three robust subtypes (k = 3), with a significant increase of clustering stability from k = 2 to 6 (Figure 1A and Appendix A). As we observed a marginal increase of clustering stability after k = 3 (less than 0.2) (Appendix A), we clustered and defined these three consensus clusters as Subtype 1, Subtype 2 and Subtype 3. Cluster significance was evaluated by SigClust. The boundary between Subtype 1 and Subtype 2 was statistically significant. However, no significance was observed between Subtype 1 versus Subtype 3 and Subtype 2 versus Subtype 3, suggesting that partial m^6^A regulators are likely activated in Subtype 1 and 2 (Appendix A). PCA also revealed a separation of Subtype 1 and Subtype 2 and an overlap of Subtype 3 (Figure 1B). Samples in each subtype were identified based on their positive silhouette width, with few exceptions, suggesting that a higher similarity was observed to their own cluster than to any other clusters (Figure 1C). We also observed a difference of m^6^A regulator expression between subtypes (Figure 1D), where half of the m^6^A regulators were highly expressed in Subtype 1 and Subtype 2. Notably, almost all m^6^A regulators were highly expressed in Subtype 3, suggesting an extra-active m^6^A modification likely occurred in Subtype 3 of PCa.

### 3.2. Different Immune Responses and the Activation of Metabolic Pathways Were Observed in m^6^A Regulator Subtypes of PCa

Since PCa revealed three distinct subtypes in the expression of m^6^A regulators, we further explored the differentially expressed genes between subtypes to uncover the key pathways mediated by m^6^A regulators. A DEG analysis in pairwise fashion identified 24 genes that were significantly expressed between the three m^6^A subtypes (Appendix A), suggesting the mRNAs from these 24 genes were likely modified or co-regulated by m^6^A regulators. It is notable that all samples clustered in Subtype 3 had an intermediate or high risk of Gleason scores, while Subtype 1 and 2 had mixtures of PCa patients with low, intermediate and high Gleason scores (Appendix A), suggesting that m^6^A regulation likely occurred in all stages of PCa but was more pro-active in progressive PCa. Through the Gene Ontology and pathway analysis in Metascape, we found that these differentially expressed genes were enriched in metabolic processes and the immune response (Figure 2A). As advanced PCa demonstrated an overwhelming de novo resistance to immune checkpoint blockade [34], an investigation on the relationship of m^6^A regulators and immune cell infiltration was conducted. To explore the immune cell infiltration patterns in each subtype, we performed a prediction for immune cell proportions by CIBERSORT. Notably, the levels of CD8+ T cells, CD4 memory T cells and Follicular helper T cells, and activated dendritic cells were markedly different in each subtype (ANOVA, *p* < 0.001, respectively) (Figure 2B), suggesting that m^6^A expression levels may contribute to the immune cell (T cell) infiltration in PCa. A relatively low T cell infiltration was observed in Subtype 2, indicating Subtype 2 was likely an “immune desert” tumor, and immunotherapy would probably be ineffective in this subtype. To confirm the enriched key pathways between subtypes, we then performed the GSEA on these 454 primary PCa samples. The GSEA confirmed that the oxidative phosphorylation and fatty acid metabolism were enriched in the Subtype 1 PCa, while the inflammatory response pathway was downregulated in Subtype 3 (Appendix A). These results reveal that progressive PCas with a high level of m^6^A regulators likely have distinct biological pathway activations.

### 3.3. Clustering of m^6^A Regulators Was Associated with Progression in PCa

To test whether the clustering of m^6^A regulators was associated with prostate patient outcomes, we compared the survivals in each subtype. Interestingly, there were no significant differences in overall survival across all m^6^A subtypes (*p* = 0.7, Log-rank test), but patients in Subtype 3 had a worse recurrent-free survival (*p* = 0.02, Log-rank test) (Figure 3A), suggesting that PCa recurrence was likely modulated by m^6^A regulators. Notably, the prognostic capacity of m^6^A regulator clustering was more significant in T3 PCa patients than in T1 and T2 PCa (Appendix A). Considering that a few m^6^A regulators were differentially expressed between 55 normal prostate tissues and 454 prostate cancers (Appendix A) and PCa metastasis was a more critical event than recurrence, we then compared the m^6^A expression in the GSE147493 dataset, which included PCas with or without metastasis. Seven of 21 m^6^A regulators (HNRNPA2B1, FMR1, METTL14, KIAA1429, YTHDF1, ALKBH5, HNRNPC) displayed different expression patterns between 62 metastatic and 37 non-metastatic PCas (Figure 3B), suggesting that these seven m^6^A regulators were associated with PCa metastasis. More importantly, we also observed that the subtypes clustered by m^6^A regulators were associated with Gleason Scores. Notably, no PCa in Subtype 3 had a tumor with a Gleason Score less than 7 (Appendix A). Considering that cancer stem cells play a critical role in cancer initiation and the origin of cancer metastasis in PCa, we then further tested the Stemness Index between subtypes. As we expected, Subtype 3 had the significantly highest Stemness Index of all three subtypes (Appendix A). These results suggest that clustering of PCa by the expression of m^6^A regulators was associated with PCa progressive phenotypes (including recurrence and metastasis).

### 3.4. m^6^A Score Could Be a Surrogate Marker for m^6^A Activation in PCa

To generalize m^6^A subtyping broadly and directly predict the metastasis of PCa, we established an m^6^A score to quantify the activation of m^6^A in PCa. Notably, seven m^6^A regulators were also differentially expressed between 454 PCas and 55 normal prostate tissues (Appendix A), suggesting these seven m^6^A regulators likely orchestrated the metastasis by activation of m^6^A in PCa. Therefore, we took these seven m^6^A regulators as the key signature to investigate their expression in PCa. In brief, the m^6^A score of each sample was the sum of the first and second Principal Component in the given sample. To test that our m^6^A scores could surrogate the subtypes of PCa, we first compared the m^6^A scores between subtypes. As expected, the three m^6^A subtypes had different expressions of m^6^A scores, and Subtype 3 had the lowest level of m^6^A score among all subtypes (Figure 4A). The m^6^A score had an inverse correlation with most m^6^A regulators (Figure 4B), suggesting that the m^6^A score was likely a comprehensive surrogate for m^6^A inactivation in PCa. In addition, the m^6^A score also had a significant correlation with the Gleason score (Figure 4C), suggesting the m^6^A score could be a surrogate for PCa progression. A previous study revealed that the RNA demethylase ALKBH5 was selectively activated in cancer stem cells and promoted tumorigenesis in leukemia; therefore, we tested whether our m^6^A score was associated with cancer stem cells in PCa [35]. To further elucidate the concordance of the m^6^A score and stemness in PCa, we also inferred the proportion of cancer stem cells in PCa using the Stemness Index and found that the m^6^A score was significantly associated with the Stemness Index (r = −0.362, *p* = 9 × 10^−7^) (Figure 4D) [33]. Taken together, these results suggest that the m^6^A score could be a promising predictive biomarker for PCa metastasis.

To further validate the progression prediction for PCa by m^6^A score, we compared the m^6^A score between PCas with metastasis and non-metastasis. As expected, metastatic PCas had lower levels of m^6^A scores than PCas without metastasis (Figure 5A). Hence, we conducted a ROC to investigate the predictive capacity of the m^6^A score on PCa metastasis. The AUC of the m^6^A score for the prediction of metastatic PCa was 0.633 (Figure 5B). To further test the predictive effectiveness of them^6^A score, we then performed the m^6^A score for another independent cohort (GSE6919) that contained 25 metastatic PCas and 60 primary PCas. As expected, our m^6^A score displayed a significant discrimination of metastatic PCas from primary PCa, and the predictive accuracy in the validation cohort was 89.5% (Figure 5C,D). Our results demonstrate a superior diagnostic capacity of metastatic PCa to SNPH (which is a biomarker for metastatic PCa [36]) (Appendix A), suggesting that the m^6^A score is a promising diagnostic biomarker for metastatic PCa.

Lastly, we tested the genetic and epigenetic alterations of all m^6^A regulators in PCa because these alterations were considered the common events in other types of cancers. Within 331 PCa samples that underwent whole-genome sequencing, only 19 (5.74%) patients had genetic variation, and ZC3H13 was the most common alteration of all m^6^A regulators. Half of the m^6^A regulators (WTAP, ALKBH5, YTHDC1, YTHDF1, YTHDF2, YTHDF3, RNPA2B1, HNPRNPC, FMR1, ELAVL1) did not have any genetic alteration (Appendix A). We then explored whether the DNA methylation profiles of these m^6^A regulators had changed the expression of m^6^A regulators. We selected the CpG loci located in the promoter areas of the m^6^A regulators to compare their methylation levels. Similar to the genetic profiles, the DNA methylation levels were consistent across all PCa in m^6^A regulators (Appendix A). As overexpression of ERG and SPOP mutation are associated with PCa progression and proliferation, we also investigated the concordance of our m^6^A subtyping with TCGA taxology subtypes and SPOP mutation status. The majority of m^6^A subtype PCas consisted of the “1-ERG” subtype, confirming that m^6^A subtype 3 PCa is likely associated with PCa metastasis because metastatic PCa is characterized by TMPRSS2-ERG fusion. It also indicated ERG probably coordinated m^6^A activation to promote PCa metastasis. We then examined ERG gene expression in TCGA molecular classification and m^6^A subtyping and found that m^6^A subtype 3 had the highest gene expression of ERG across all subtypes, as “1-ERG” did. In contrast, we did not conclude the association of SPOP mutation with m^6^A subtype 3, as all m^6^A subtype 3 came from SPOP wildtype PCa (Appendix A). These results suggest that m^6^A regulators likely participate in PCa metastasis via ERG expression rather than SPOP mutation, though we understand that it requires further experiments to validate. Taken together, these results suggest that m^6^A regulators were not altered by genetic alteration or methylation status in PCa.

## 4. Discussion

Second to the skin cancer, PCa is the most prevalent type of carcinoma afflicting men in North America. PCa is also the second leading cause of cancer death worldwide. Androgens regulate the growth of normal and malignant prostate tissue through the action of the AR in both epithelial and stromal cells. Hence, ADT is foundational in the management of metastatic PCa, and this treatment is effective in managing this disease temporarily for the majority of patients. However, cancer cells become castration-resistant, eventually leading to disease progression to mCRPC. In disseminated phases, especially with mCRPC upon ADT failure, these patients have poor survival rates. A few studies uncovering the heterogeneity of mCRPC beyond the somatic mutation highlights the importance of transcriptome alteration during ADT resistance [37,38]. In this study, we demonstrated that the subtypes of PCa according to the clustering of m^6^A regulators of expression are significantly associated with PCa metastasis. M^6^A regulators likely contribute to the heterogeneity of PCa metastasis through the key genes involved in immune cell infiltration and metabolism. 

Like other types of cancer, m^6^A regulators are associated with patients’ recurrent-free survival in primary PCa. We performed a survival analysis of PCa to explore the correlation of survival with m^6^A subtypes. Interestingly, there is no statistical significance between m^6^A subtypes on overall survival, but there is in recurrence-free survival. This is likely because most primary PCas are biologically slow-growing and the regulatory effects mediated by m^6^A regulators are marginal for primary PCa. Considering the DEGs of m^6^A subtypes enriched in the metabolic pathway are important to PCa metastasis, we then investigated the correlation of m^6^A regulator expression with metastasis. 

The evidence from recent studies has defined that PCa displays dynamic patterns of evolution in the context of ADT-associated metastasis [39]. There are two common mechanisms of metastasis-to-metastasis spread. First, in a process named ‘cross-metastatic seeding’, subclones within a metastasis can come from another metastatic site instead of the primary neoplasm [40], which was also confirmed in response to therapy in a PCa patient [41]. Second, in a process called ‘polyclonal seeding’, the same sets of subclones can seed multiple sites of metastasis [40]. The polyclonal seeds may share multiple subclones for two or more metastases, which indicates that these subclones might cooperate in function to promote metastatic progression. The phenomenon that distant metastases could also reseed the surgical bed provides evidence that makes use of pre-existing supportive niches [41]. This process of ‘tumor self-seeding’, which has previously been observed with PCa CTCs, can accelerate tumor growth [42].

Prostate tumors are defined ranging from indolent to highly aggressive in clinical settings. Most of these tumors are localized and treated based on their stage or Gleason score, which is likely evaluated via invasive biopsy. At the first diagnosis, the majority of these tumors are identified as advanced or aggressive and often are accompanied by micro-metastasis at secondary locations. Among the different sites in the body, PCa has a high propensity for metastasizing to the bone [43]. Aside from visceral metastasis, the presence of skeletal metastasis in PCa patients is related to the remarkably lower overall survival of 14 months [43], signifying the metastatic locations are critical determinants of disease prognosis.

A few attempts have been made to predict metastatic PCa from non-metastatic primary PCa but there is no consensus regarding a reliable biomarker yet [44,45,46,47,48]. In addition, other studies fail to demonstrate the relationship of the expression of the m^6^A regulator with PCa metastasis, though they can indicate patients with PCa outcomes [49,50]. Compared to the overall survival of all PCas, prediction of metastasis would be more helpful in the clinical practice, as most primary PCa is localized. Therefore, through applying consensus clustering of gene expression, we can predict the cancer metastasis more accurately than through conventional histopathology [51]. Combined with other parameters such as clinical information and MRI, our score could help establish a strategy for PCa to inform clinicians on whether to escalate or deescalate the current treatment, though it requires validation in a multi-center prospective clinical trial [8,10]. Our m^6^A score can differentiate metastatic PCa from non-metastatic PCa, though highlighting its clinical utility though a large-scale clinical validation cohort is still required to compare with other biomarkers such as AR-V7 [52]. Further investigation on the association of m^6^A regulator expression with multiple metabolic pathways in the development of PCa metastasis is also needed [53].

As mediators of post-transcriptional modification, m^6^A regulators participate in a variety of biological processes, including cancer tumorigenesis and progression. METTL3 is one of the well-characterized m^6^A regulators in PCa. Cai et al. first reported that METTL3 promoted cell growth in PCa by modifying the mRNA of GLI1, the key nuclear mediator in the Hedgehog pathway [54]. High expression of METTL3 is also associated with bone metastasis in PCa. By transferring the methyl group to the mRNA of ITGB1, METTL3 enhances PCa cell motility to accommodate bone metastasis [55]. In addition, PCa progression is significantly associated with METTL3, suggesting that METTL3 is likely a key regulator in PCa.

More attention has been paid to the fact that m^6^A mediates immune cell infiltration during cancer growth and metastasis. More immune cell infiltration into tumors was observed in the reduction of METTL3, which was activated by NF-kB and ERK pathways [56]. Su et al. also found that the FTO inhibitor significantly suppresses cancer stem cell maintenance and immune checkpoint gene expression, highlighting the potential efficiency of the m6A regulator inhibitor [57]. In PCa, immune cell status could surrogate the treatment efficacy [58]. Our results show the difference of immune cell infiltration in different subtypes, highlighting the capacity of stratification for patients according to m^6^A score in the future.

A few existing scores for PCa have demonstrated their clinical utilities [59]. D’Amico classification is used for predicting the risk of treatment failure in 5 years. Lack of multiple risk factors limits its accuracy. A nomogram-based scores such as the Kattan nomogram are also developed for PCa, but it is sometime hard for the clinician to interpret the results [60]. The CAPRAS score is a straightforward scoring tool for predicting PCa patients’ outcomes; however, it is only built for prediction after a range of treatment strategies [61]. Most of them are built upon clinical information (including age, clinical stage and Gleason Score) but none of them include any biological parameters other than the serum PSA level. Therefore, it is worth investigating the combined prediction effectiveness of these scores with the m^6^A score in the future.

Although we established a comprehensive landscape of m^6^A expression patterns in PCa and validated the predictive capacity of our m^6^A score in PCa metastasis, there are some limitations that need to be addressed. First, the m^6^A score we constructed should be tested in more independent cohorts to verify its clinical utility. We intend to validate our m^6^A score in specimen samples in the future. Second, the m^6^A score is a surrogate for the comprehensive activation of m^6^A regulators, but the most critical molecule(s) in RNA modification have not been fully investigated in PCa. The detailed mechanism of m^6^A regulators and the relationship with their target genes have not yet been elucidated. In addition, the regulatory mechanism of m^6^A regulators and lncRNA expression has remained unclear [31]. Lastly, the driving force of the differential expression pattern of m^6^A regulators beyond DNA methylation and genomic alteration should also be further investigated in metastatic PCa.

## 5. Conclusions

In conclusion, the clustering of m^6^A regulators in PCa reveals the heterogeneity of PCa, and one subtype (m^6^A Subtype 3 PCa) is significantly associated with PCa metastasis. The gene signatures in this “metastatic” PCa are enriched in the immune cell response and metabolic response. Our m^6^A score is potentially a biomarker for metastatic PCa.

## Figures and Tables

**Figure 1 cancers-14-04035-f001:**
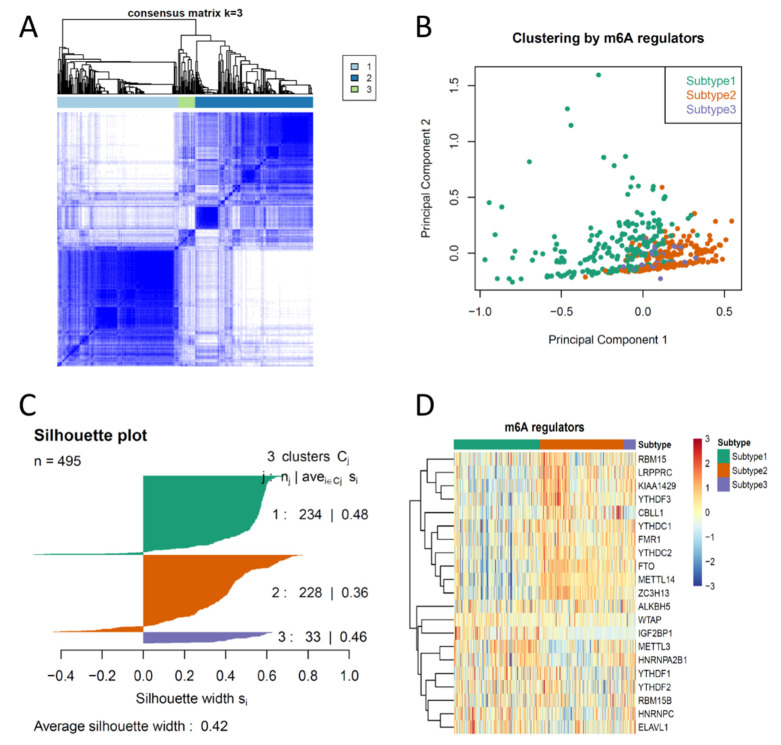
Consensus clustering of m^6^A regulators demonstrates three distinct subtypes in prostate cancer (PCa). (**A**) Heatmap showing the consensus matrix (k = 3) of m^6^A regulator expression. (**B**) PCA showing the clustering of PCa by m^6^A regulators. (**C**) Silhouette plot showing the similarity of samples clustered in sample subtypes. (**D**) Heatmap showing the distinct pattern of each m^6^A regulator in all 454 PCa samples.

**Figure 2 cancers-14-04035-f002:**
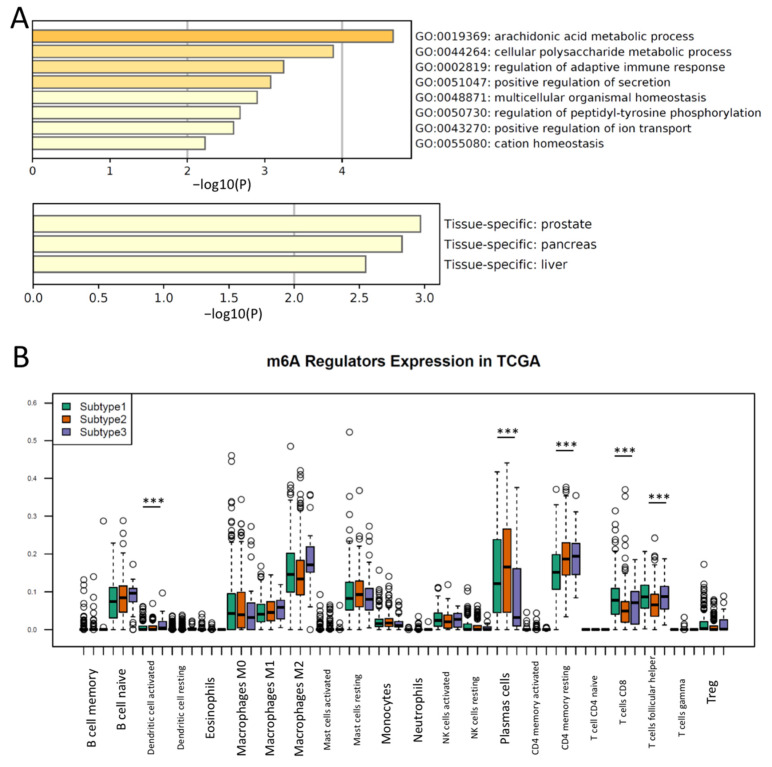
Different responses of immune cells are enriched in the three subtypes of PCa. (**A**) Pathways of DEG between m^6^A subtypes are enriched in the metabolic process and immune response (upper). The subtype-specific DEGs are enriched in PCa (lower). (**B**) Boxplot showing the different immune cell infiltrations between m^6^A subtypes. ANOVA test; ***, *p* < 0.0001.

**Figure 3 cancers-14-04035-f003:**
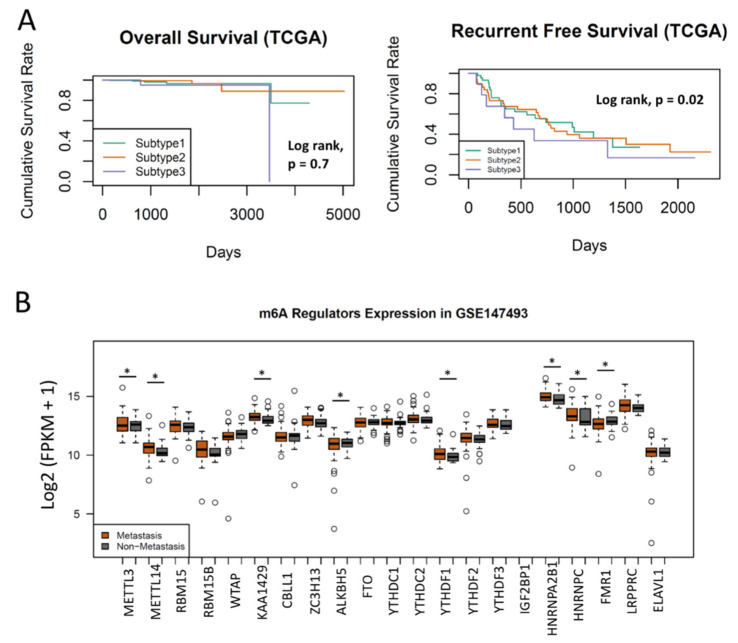
M6A in recurrence and metastasis of PCa. (**A**) Subtyping of PCa by m^6^A regulators is not associated with patients’ overall and recurrent-free survival (Log rank test, *p* = 0.7 and 0.02, respectively). (**B**) m^6^A regulators are associated with PCa metastasis. Seven m^6^A regulators are differentially expressed between metastatic and non-metastatic PCa. Student’s *t* test. * *p* < 0.05.

**Figure 4 cancers-14-04035-f004:**
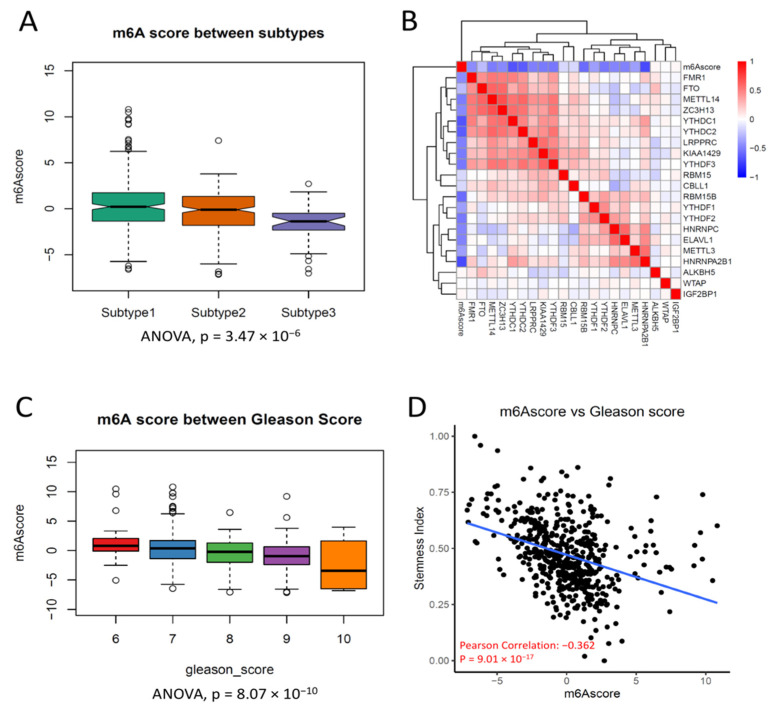
The m^6^A score is related to the Gleason Score and the stemness score. (**A**) Boxplot showing the different m^6^A scores between subtypes of m^6^A. Subtype 3 has the lowest m^6^A score. ANOVA, *p* = 3.47 × 10^−6^. (**B**) Heatmap of the correlation between m^6^A regulators. The correlation coefficient is highlighted by color. (**C**) Boxplot showing that the m^6^A score is inversely associated with the Gleason score (ANOVA, *p* = 8.07 × 10^−10^). PCa with a higher Gleason Score has a lower m^6^A score. (**D**) m^6^A score is inversely correlated with the Stemness Index (Pearson Correlation, r = −0.362; *p* = 9.01 × 10^−17^).

**Figure 5 cancers-14-04035-f005:**
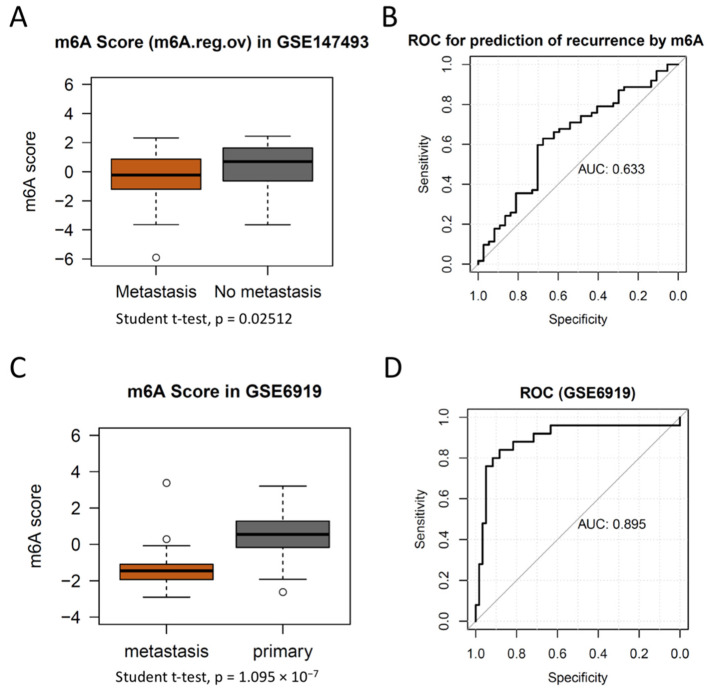
The m^6^A regulators are differentially expressed between metastatic and non-metastatic PCas. (**A**) Boxplot showing metastatic PCa has a significantly lower m^6^A score. Student *t* test, *p* = 0.02512. (**B**) The m^6^A score showed a moderate discrimination of metastatic PCa from non-metastasis. The accuracy of the m^6^A score is determined by the Area Under the Curve (AUC). The effectiveness of the m^6^A score is validated in another independent cohort (GSE6919). The boxplot (**C**) showing the metastatic PCa has a significantly lower expression of m^6^A score than primary PCa. (**D**) ROC of the GSE6919 dataset demonstrating the prediction accuracy for metastatic PCa is 89.5% in the validation cohort.

## Data Availability

The raw data will be provided by the corresponding authors once reasonable request. The R code for this study is released at https://github.com/mackaay/PCa_m6A (accessed on 8 October 2020).

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
