# Peer review of "m6A Methylation Regulators Are Predictive Biomarkers for Tumour Metastasis in Prostate Cancer"

_cancers, 2022, doi:10.3390/cancers14164035_

Round 1
Reviewer 1 Report
This work aims to find an association between some expression patterns of m6A regulators and the propensity to metastasize of the prostate cancer (PCa). m6A is the most common post-trascriptional mRNA modification and it’s regulated by three kinds of regulators: writers (methiltrasferase), erasers (demethylase) and readers (binding proteins); an association has been noted between the expression pattern of these regulators and the aggressiveness of the cancer in terms of methastasis. The authors first identified three clusters in PCa based on the expression profile of 21 m6A regulators, then they analyzed before the overall survival difference between this three clusters, finding no significative difference, then they analyzed the propensity to metastasize of the PCa relying on this three clusters and they found out that m6A Subtype 3 PCa is significantly associated with PCa Methastasis.
My suggestions:
English language should be simplified, the writing is not very clear in some points
The introduction of the work should explain more precisely the indications for the various therapies because with PCa we can use a lot of effective drugs and strategies, so it’s important to avoid wrong therapies and overtreatment. I can suggest to analize this work about this topic: https://pubmed.ncbi.nlm.nih.gov/34572950/
m6A score should be explained with more clarity in the article and it should be validated by a large scale trial in order to use it in clinical practice
A reflection should be done on the behaviour to be adopted once identified a dangerous pattern in a PCa. Authors should deepen what to do in the clinical practice if a patient express a high risk pattern, trying to avoid overtreatment or vice versa, to underestimate the risk
The correlation between the expression of m6A regulators and the tumor immune cell infiltration should be deepen by the authors because this could be a hot point for the PCa therapy, specially in some particular patient conditions where cancer therapy should be stopped not to worsen the immunological situation, for example during COVID-19 infection. m6A score could be an important item to define the aggressivness of the cancer, making the difference between clinical situations where therapies can be temporarily suspended or not. At this regard i can suggest the reading of this work: https://pubmed.ncbi.nlm.nih.gov/32570240/
No mention in this work is done about the costs or the availability of the strumentation to sequence the RNA or the m6A related genes. This could be a big limitation of this strategy because of its elevated cost and the non-accessibility for all patients affected by PCa.
Author Response
We appreciate the comments made by this reviewer.
English language should be simplified, the writing is not very clear in some points
We have read through this manuscript again and made a few amendments on it to make it more readable and precise. We also have an English editing service (AJE) to help us the language (verification code 9202-7F9C-F571-D5F8-8E6A).
The introduction of the work should explain more precisely the indications for the various therapies because with PCa we can use a lot of effective drugs and strategies, so it’s important to avoid wrong therapies and overtreatment. I can suggest to analize this work about this topic: https://pubmed.ncbi.nlm.nih.gov/34572950/
We agree with the reviewer’s comments and this reference is really helpful to explaining the therapeutic selection in the clinical practice. We have added the corresponding context in the revised manuscript.
m6A score should be explained with more clarity in the article and it should be validated by a large scale trial in order to use it in clinical practice
We added a few more words regarding m6A to clarify its potential function in the cancers. Due to the limitation of resource we have, we could be able to access our m6A score in a large clinical trial for now but we have set a plan to run it in the near future. We also acknowledge it as one of our limitations.
A reflection should be done on the behaviour to be adopted once identified a dangerous pattern in a PCa. Authors should deepen what to do in the clinical practice if a patient express a high risk pattern, trying to avoid overtreatment or vice versa, to underestimate the risk
The strategy for PCa is critical in the clinical practice. We added a few more words around it in the Discussion session to show it might be helpful in the clinical decision of escalating the treatment for PCa along with other tools.
The correlation between the expression of m6A regulators and the tumor immune cell infiltration should be deepen by the authors because this could be a hot point for the PCa therapy, specially in some particular patient conditions where cancer therapy should be stopped not to worsen the immunological situation, for example during COVID-19 infection. m6A score could be an important item to define the aggressivness of the cancer, making the difference between clinical situations where therapies can be temporarily suspended or not. At this regard i can suggest the reading of this work: https://pubmed.ncbi.nlm.nih.gov/32570240/
We agree with the reviewer’s comment that immune response in PCa should be considered. Numerous studies have demonstrated that immune response plays a critical role in PCa development, metastasis and the efficacy of therapeutics. We have added a few sentence discussing it in the revised manuscript. This reference has also been cited in this manuscript.
No mention in this work is done about the costs or the availability of the strumentation to sequence the RNA or the m6A related genes. This could be a big limitation of this strategy because of its elevated cost and the non-accessibility for all patients affected by PCa.
As it is a proof-of-concept study, main aim of this study is to demonstrate the significance of m6A in PCa. We could not be able to perform a cost or health economic analysis in this study yet. All authors agree that cost-effect analysis should be taken into consideration if we would like to apply this into the clinical practice. Sequencing including targeted sequencing for m6A might not be affordable in all PCa. Instead, we are thinking about PCR or IHC for m6A regulator testing in the future.
Reviewer 2 Report
Under Results, Section 3.1, regarding the statement "Notably, almost all m6Aregulators were highly expressing Sutype 3, suggesting an extra-active m6A modification likely occurred in Subtype 3 of PCa" -- this interpretation is hard to see, i.e., it is not clearly visible in the graphics shown in 1D. Can the visual distinctions be improved. Alternatively, perhaps the authors should include a calculation of the % of elements under Subtype 3 that are over-expressed, and compare that to the % of elements over-expressed in the other two subtypes.
Under Section 3.2, the authors comment on levels of CD8+ T cells and activated dendritic cells. Think they should also comment on the other categories of immunocytes that showed significant differences between subtypes (i.e., CD4 memory T cells and Follicular helper T cells) and what these differences might suggest.
Under Sect 3.3, line 246, the phrase "recurrent free survival" is more usually written as "recurrence-free survival". Think authors should use this throughout manuscript and in figures (e.g., Figure 3A).
Under Section 3.4, lines 288-291, the inference process used to determine the proportion of cancer stem cells, needs to be described in more detail. For instance, how was machine learning in reference #31 employed?
Some aspects of Figure 5 don't make sense: In Panel A, what does the one open circle represent? Also in Panel A, the two box plots do not seem very different; how was a p value of 0.025 derived? In Panel C, what do the 3 open circles represent?
Under Discussion, line 396, "Comparing with" should be "Compared to".
Also under Discussion, some remarks about how the m6A score compares to, or might be incorporated in, existing nomograms for prostate cancer risk stratification, such as the D'Amico classification system, the CAPRA score, or Kattan nomogram.
Author Response
We appreciate the comments made by this reviewer.
Under Results, Section 3.1, regarding the statement "Notably, almost all m6Aregulators were highly expressing Sutype 3, suggesting an extra-active m6A modification likely occurred in Subtype 3 of PCa" -- this interpretation is hard to see, i.e., it is not clearly visible in the graphics shown in 1D. Can the visual distinctions be improved. Alternatively, perhaps the authors should include a calculation of the % of elements under Subtype 3 that are over-expressed, and compare that to the % of elements over-expressed in the other two subtypes.
We highlighted the relatively highly expressed m6A regulators in black boxes in Figure 1D. That would help indicate the m6A expression pattern in each subtype.
Under Section 3.2, the authors comment on levels of CD8+ T cells and activated dendritic cells. Think they should also comment on the other categories of immunocytes that showed significant differences between subtypes (i.e., CD4 memory T cells and Follicular helper T cells) and what these differences might suggest.
We are sorry for this mistake. Now we have added CD4 memory T cells and Follicular helper T cells into our context. We also added a few words to show the indication of this result. A paragraph is also added in the Discussion.
Under Sect 3.3, line 246, the phrase "recurrent free survival" is more usually written as "recurrence-free survival". Think authors should use this throughout manuscript and in figures (e.g., Figure 3A).
We appreciate the reviewer’s suggestion. We have made the changes correspondingly.
Under Section 3.4, lines 288-291, the inference process used to determine the proportion of cancer stem cells, needs to be described in more detail. For instance, how was machine learning in reference #31 employed?
We added a paragraph in the Method (section 2.6) to illustrate how is this algorithm developed. Briefly, Malta et al identified 24 stem cell signatures and they can predict the stemness in a given sample based on the expression of this 24-gene signature.
Some aspects of Figure 5 don't make sense: In Panel A, what does the one open circle represent? Also in Panel A, the two box plots do not seem very different; how was a p value of 0.025 derived? In Panel C, what do the 3 open circles represent?
We are sorry for the misunderstanding. The open circles in the boxplots represent the outliers in the dataset, which means extremely high or low m6A scores in the groups. The P value was determined by the student’s t test in the R environment.
Under Discussion, line 396, "Comparing with" should be "Compared to".
Thank the reviewer’s suggestion and we had made the amendment accordingly.
Also under Discussion, some remarks about how the m6A score compares to, or might be incorporated in, existing nomograms for prostate cancer risk stratification, such as the D'Amico classification system, the CAPRA score, or Kattan nomogram.
We appreciate the reviewer’s suggestion, and it is really useful. We added a few remarks regarding these risk prediction tools in the Discussion. Each tool has its own advantages and disadvantages. We have summarized them in the revised manuscript as well.
Reviewer 3 Report
The manuscript by Liang et. al. studied the effect the m6A RNA methylation on prostate cancer carcinogenesis and metastasis using RNA seq data from TCGA. Based on expression of 21 methylation regulator genes, they grouped 494 cases into three categories and compared oncological outcome. In addition, a m6A score was developed and was shown to predict prostate cancer metastasis. It is a novel study exploiting the role of RNA modification in prostate cancer prognostication. The study is well designed, and the results are clearly presented.
Minor comments
1. Figure 1C - n = 495 but only 454 prostate cancer samples were used in analysis per text. Please explain the discrepancy.
2. Figure 3 is titled "regulators of m6A are more relevant to metastasis than recurrence" however the figures does not support this statement. There was no direct comparison between metastasis and recurrence.
3. Line 364, "recurrent survival" should be "recurrent free survival"
4. The first paragraph of "introduction" should have the same font size
Author Response
We appreciate the comments made by this reviewer.
Minor comments
- Figure 1C - n = 495 but only 454 prostate cancer samples were used in analysis per text. Please explain the discrepancy.
We are sorry for this uncareful mistake. The total number of this study is 495 and all corresponding errors have been corrected.
- Figure 3 is titled "regulators of m6A are more relevant to metastasis than recurrence" however the figures does not support this statement. There was no direct comparison between metastasis and recurrence.
We agree with this reviewer that the previous title is not appropriate. We have changed it into “M6A in recurrence and metastasis of PCa”.
- Line 364, "recurrent survival" should be "recurrent free survival"
Thank the reviewer’s suggestion and we have corrected this error.
- The first paragraph of "introduction" should have the same font size
We thank the reviewer’s suggestion and we have change them into the same font size.
Round 2
Reviewer 1 Report
The authors answered all comments and suggestions.